# Human Exposure to Pesticides in Dust from Two Agricultural Sites in South Africa

**DOI:** 10.3390/toxics10100629

**Published:** 2022-10-21

**Authors:** Céline Degrendele, Roman Prokeš, Petr Šenk, Simona Rozárka Jílková, Jiří Kohoutek, Lisa Melymuk, Petra Přibylová, Mohamed Aqiel Dalvie, Martin Röösli, Jana Klánová, Samuel Fuhrimann

**Affiliations:** 1RECETOX, Faculty of Science, Masaryk University, 625 00 Brno, Czech Republic; 2Aix-Marseille University, CNRS, LCE, 13003 Marseille, France; 3Global Change Research Institute of the Czech Academy of Sciences, 603 00 Brno, Czech Republic; 4Centre for Environmental and Occupational Health Research, School of Public Health and Family Medicine, University of Cape Town, Cape Town 7925, South Africa; 5University of Basel, 4002 Basel, Switzerland; 6Swiss Tropical and Public Health Institute (Swiss TPH), 4002 Basel, Switzerland; 7Institute for Risk Assessment Sciences (IRAS), Utrecht University, 3584 Utrecht, The Netherlands

**Keywords:** plant protection products, residential exposure, agriculture, Africa, exposure pathway, intake dose, temporal variations, spatial variations

## Abstract

Over the last decades, concern has arisen worldwide about the negative impacts of pesticides on the environment and human health. Exposure via dust ingestion is important for many chemicals but poorly characterized for pesticides, particularly in Africa. We investigated the spatial and temporal variations of 30 pesticides in dust and estimated the human exposure via dust ingestion, which was compared to inhalation and soil ingestion. Indoor dust samples were collected from thirty-eight households and two schools located in two agricultural regions in South Africa and were analyzed using high-performance liquid chromatography coupled to tandem mass spectrometry. We found 10 pesticides in dust, with chlorpyrifos, terbuthylazine, carbaryl, diazinon, carbendazim, and tebuconazole quantified in >50% of the samples. Over seven days, no significant temporal variations in the dust levels of individual pesticides were found. Significant spatial variations were observed for some pesticides, highlighting the importance of proximity to agricultural fields or of indoor pesticide use. For five out of the nineteen pesticides quantified in dust, air, or soil (i.e., carbendazim, chlorpyrifos, diazinon, diuron and propiconazole), human intake via dust ingestion was important (>10%) compared to inhalation or soil ingestion. Dust ingestion should therefore be considered in future human exposure assessment to pesticides.

## 1. Introduction

Pesticides are the only chemicals that have been synthetized for about 70 years for their toxic properties. Their use has increased on a global scale, from 2.3 to 4.2 million tons between 1990 and 2019 [1]. In addition to their toxicity to target organisms, many pesticides cause a wide range of adverse effects for mammals, including humans [2,3]. For example, long-term exposure to some organophosphate insecticides has been associated with neurotoxic and developmental effects for chlorpyrifos [4], while diazinon exposure has induced oxidative stress, immune disorders, and gut microbiota dysbiosis [5], and exposure to several pesticides cause DNA damage [6,7]. Furthermore, there has been significant evidence of pesticide contamination of several environmental matrices, such as air, soil, or water [8,9,10,11,12,13,14]. Consequently, many concerns have arisen worldwide about the negative impacts of pesticides on the environment and human health. In particular, agricultural region residents are a highly exposed population, with higher quantification frequencies and levels of pesticides found in various environmental media from rural areas, such as air [15,16], dust [17,18,19], silicone wristbands [20,21], and human samples such as urine [22,23,24] and blood [25,26]. In addition, several epidemiological studies have found an association between residential proximity to agricultural fields and diseases such as autism [27,28], Parkinson’s disease [29], childhood cancer [30,31], and also with neurobehavioral effects [32].

Humans are exposed to pesticides via three pathways: (i) ingestion of food, dust, and soil; (ii) inhalation; and (iii) dermal contact with products or materials containing pesticides [33]. For many legacy chemicals, food ingestion is generally the major contributor (>90%) to human exposure [34,35] although some studies have reported that inhalation [36] and dust ingestion [37] could also dominate the overall exposure. For pesticides that are currently used, the contribution of each pathway to the overall exposure is not well-understood, and contradictory results have been found [34,38,39,40]. However, a clear understanding of exposure pathways is crucial to assess the pesticide exposome [41,42], which has been a growing area of research in recent years [43].

The indoor environment, where humans spend about 90% of their time [44,45], is particularly important in terms of non-dietary exposure to pesticides or other pollutants [46,47]. Pesticides can penetrate indoors in four manners. Firstly, after their application to agricultural areas, depending on the meteorological conditions and the type of equipment used, up to 30% of pesticides do not reach their targets but are transported via the air to the surrounding environment [48], a phenomenon known as spray drift [49]. In addition, several weeks after outdoor application, pesticides can be transferred to the air (secondary drift) via volatilization from soils and plants [50,51] and wind erosion of soil particles on which pesticides are sorbed, followed by dispersion [42,52]. These pesticides present in outdoor air can then infiltrate into indoor spaces via ventilation. Secondly, agricultural workers (pesticide applicators, farm workers) can also bring pesticides indoors via shoes, clothes, skin, or hair, also known as the take-home exposure pathway [53,54,55,56]. Thirdly, pesticides can be directly applied indoors against insects (e.g., mosquitoes, fleas, ticks) [17,57]. Fourth, pesticides can volatilize from products present indoors containing pesticides (e.g., wooden furniture, textiles, carpets) [58]. Given this diversity of possible sources, it has often been difficult to identify the major source of pesticides present indoors [54,59].

Dust is considered a marker of indoor pollution by organic compounds [44,47]. The pesticides levels in indoor dust are affected by several factors, including their emissions (indoor), rates of air transport from outdoor to indoor, outdoor soil brought in by shoes, removal rates by ventilation and cleaning, indoor activities, and rates of degradation indoors [33,47]. However, infiltration from outdoors seems to be the most dominant source, as many studies have shown that humans living in proximity to agricultural fields had higher levels in household dust compared to non-agricultural residents [18,42,54,55,60]. On the global scale, the many studies performed earlier on pesticides in dust largely focused on insecticides and more particularly on organophosphates and pyrethroids, while data for fungicides and herbicides are still very limited [54]. In addition, the temporal variations of pesticide levels in dust have mainly been investigated from a seasonal perspective [61,62], while only two studies have focused on shorter time scales (i.e., days) [17,63]. In order to reduce human exposure to pesticides and the related health effects for the agricultural residents, it is therefore crucial to characterize the levels of pesticides in dust and to understand their spatio-temporal variabilities.

Information on pesticide exposure of agricultural residents is extremely limited in Africa [43] and particularly using dust samples [44]. Indeed, only one study has been done in South Africa [64]. With about 26,000 tons of pesticides used on a yearly basis and about 700 active ingredients registered for agricultural use [65], this country contributes to about one-third of all pesticides used in the African continent [1] and is therefore considered as a high-risk country for pesticide pollution [66]. Unfortunately, only a few pesticides were targeted (*n* = 8) in that study, and samples were only collected at a school. Therefore, there is a real need to characterize the levels of pesticides in household dust in South Africa.

Within the larger project “Child health Agricultural Pesticide cohort study in South Africa” (CapSA), assessing the health impacts of pesticide exposure of 1000 children in South Africa [67], we have previously highlighted the presence of many pesticides in air, soil, water, and silicone wristbands of two rural agricultural areas [20,68,69,70]. In this study, we present data on pesticides quantified in dust and assess the non-dietary exposure via dust ingestion of humans living in two intensive agricultural areas in South Africa. The specific aims of this study are to (i) assess the occurrence of pesticides in dust from two agricultural areas in South Africa and its spatial and temporal variations, (ii) compare the levels of pesticides in dust between children living on farms and those living in neighboring villages, (iii) determine the human uptake of pesticides due to dust ingestion, and (iv) assess the importance of exposure via dust ingestion compared to inhalation and soil ingestion.

## 2. Materials and Methods

### 2.1. Collection of Dust Samples

The sampling campaign occurred in Western Cape, South Africa, at two different agricultural sites: Hex River Valley (33°28′ S;19°38′ E) and Grabouw (34°12′ S;19°5′ E), located about 110 km from each other (Figure 1). These two sampling sites were selected due to their intensive monoculture. At Hex River Valley, 98% of the agricultural land consists of table grapes, while at Grabouw, pome fruits are the major (81%) crop [70]. In order to characterize the highest exposure to pesticides, the sampling campaign was performed during the main pesticide application season at both sites (22 October 2018–29 October 2018 at Hex River Valley and 30 October 2018–6 November 2018 at Grabouw).

At each site, dust samples were collected in 19 individual households representing different exposure scenarios and one school. Information on the recruitment of participants is available elsewhere [20]. For each site, half of the households were located on the farms, and half were located in the nearby village, within a distance to agricultural fields of <50 m and >0.5 km, respectively. Dust samples were taken on the floor of the child’s bedroom for the households and of three classrooms for the two schools. Dust sampling was repeated after seven days at the same locations. A vacuum cleaner was used to collect dust samples, as this technique detects more compounds than wiped dust [58] or doormat dust [53]. The dust samples were collected on a quartz fiber filter (QFF, QMA, 101.6 mm, Whatman, UK) using a stainless steel inlet equipped with a pre-separation mesh sieving particles up to 1 mm connected to a conventional vacuum cleaner. This dust size fraction has been shown to contain most organic contaminants and to be relevant in terms of human exposure [71,72]. Ethanol was used to clean the sampling head of the vacuum cleaner prior to each use. The sampled surface varied from 1 to 8 m^2^. After vacuuming, dust samples were immediately folded, wrapped in aluminum foil in order to avoid sunlight degradation, and placed into a zip-lock polyethylene bag. All the samples were carried to the University of Cape Town within a cooling box at 5 °C where they were stored in a freezer at −18 °C until shipment to RECETOX, Czech Republic.

### 2.2. Sample Preparation and Chemical Analysis

From the samples collected, all those from day 7 (*n* = 41) and only 13 from day 1 were analyzed for pesticides, resulting in a total amount of 54 dust samples. About 0.1 g of the dust samples were extracted with 10 mL of methanol using an ultrasonic bath, then centrifugated for 10 min at 10,000 rcf. This extraction was done for three cycles, and the final extract volume was 30 mL. The extracts were concentrated at 40 °C with a gentle stream of nitrogen and passed through Chromafil syringe filters (nylon membrane, 25 mm diameter, pore size 0.45 µm, Machery-Nagel, Düren, Germany) into mini vials. The extract volume was brought exactly to 0.5 mL by weight, and 0.5 mL of ultrapure water (Sartorius, Göttingen, Germany) was added to have the final volume 1 mL methanol:water 1:1. All samples were analyzed using a high-performance liquid chromatograph (Agilent 1290, Agilent, Santa Clara, CA, USA) with a Phenomenex Luna C-18 endcapped analytical column (100 mm × 2.0 mm × 3 μm). Analyte detection was performed by tandem mass spectrometry using an AB Sciex Qtrap 5500 (AB Sciex, Concord, ON, Canada), operating in positive electrospray ionization (ESI+). The isotope dilution method was used to quantify the analytes. The instrumental limits of detection (iLOD) and quantification (iLOQ) were defined as the quantity of an analyte with a signal-to-noise ratio of 3:1 and 10:1, respectively. Details on the analytical method used have been described elsewhere [68,69].

In total, thirty individual pesticides, including seventeen herbicides (i.e., acetochlor, alachlor, atrazine, chlorotoluron, chlorsulfuron, dimethachlor, diuron, fluroxypyr, isoproturon, metamitron, metazachlor, metribuzin, pendimethalin, pyrazon, simazine, S-metolachlor, and terbuthylazine); nine insecticides (i.e., azinphos methyl, carbaryl, chlorpyrifos, diazinon, dimethoate, fenitrothion, malathion, parathion methyl, and pirimicarb); and four fungicides (i.e., carbendazim, prochloraz, propiconazole, and tebuconazole) were analyzed in this study. Among these 30 pesticides, 27 are registered for agricultural use in South Africa [65]. In addition, 15 of the pesticides quantified in this study are widely used on the global scale [73], and 14 have been identified as highly hazardous or high-risk pesticides [74]. Even though the amount of pesticides investigated in this study is small compared to the almost 700 pesticides authorized for agricultural use in South Africa [65], it is more than most of the previous studies focused on the residential exposure to pesticides of agricultural residents, which looked at a median of seven pesticides [43].

### 2.3. Quality Assurance and Quality Control

In this study, six field blanks and seven solvent blanks were analyzed as per samples. None of the targeted pesticides were found in those blanks, suggesting that no contamination occurred during sampling, transport, sample preparation, and analysis. Recoveries of individual pesticides, determined from spiking experiments of QFFs, ranged from 42.5% ± 2.9 (acetochlor) to 120.1% ± 1.9 (chlorsulfuron) (Appendix A).

### 2.4. Human Exposure via Dust Ingestion and Comparison with Inhalation and Soil Ingestion

The daily intakes of pesticides via dust ingestion (DI_ingestion_dust_, in pg day^−1^ kg^−1^) were estimated for children (6 to 11 years) who are more sensitive to pesticide exposure [3] and adults (>21 years) using both the median and the maximum concentrations observed for each type of site as [75]:(1)$$\mathrm{DIingestion}\_\mathrm{dust}\;=\frac{\mathrm{Cdust}\;\times \;\mathrm{IngRdust}\;\times \;\mathrm{AF}\;}{\mathrm{BW}}$$
where C_dust_ is the dust concentration (in pg g^−1^), and IngR_dust_ is the dust ingestion rate (in g day^−1^), AF is the absorption factor via dust ingestion (unitless), and BW is the body weight (in kg). All the input parameters used [75] are provided in Appendix A. Here, we assumed that the children spent 30% of their time at school and the remaining at home, while exposure for adults were estimated as if they spent all their time at home. Many studies have assumed that all pesticides present in dust were bioaccessible due to the lack of knowledge on bioaccessibility of pesticides ingested from indoor dust [47], leading to a possible overestimation of the health risks [76]. In a review, a bioaccessibility factor varying from 0.06 to 0.52 was reported for both legacy and current-use pesticides with a median of 0.14 [76], which we used in this study. We decided to not determine the human uptake via dermal contact with dust, as the exposure factors needed are associated with large uncertainties [47], and many studies for other organic pollutants have shown that this exposure pathway was several orders of magnitude lower than dust ingestion [47,77].

The pesticide daily intakes from dust ingestion from this study were compared with those from inhalation and soil ingestion previously reported from the same campaign [68]. In addition, we also estimated the health hazards due to the exposure via dust ingestion, inhalation, and soil ingestion of pesticides using (i) hazard quotients determined as the ratio of the daily intake to the acceptable daily intake via all routes of exposure obtained from European database [78], (ii) hazard index, and (iii) relative potency factors, as previously described [68].

### 2.5. Data analysis

Mann–Whitney tests were used to compare the differences between the two sampling days, the two studied areas, the two types of sites (farm vs. village), and the two population groups in terms of the dust concentrations of pesticides and human exposure. Significant differences were considered when *p*-value < 0.05. For these analysis, including summary statistics, the pesticides that were quantified in >50% of the samples were considered, and the values under LOQ were imputed with half LOQ. The software MATLAB^®^ (version R2017a) was used to perform the data analysis and create all figures except for the map (Figure 1), which was done with the software QGIS (version 3.4 Madeira).

### 2.6. Ethical Statement

Informed consent was obtained from a member of each household. The study received ethical clearance from the University of Cape Town’s Research Ethics Committee (HREC 637/2018).

## 3. Results

### 3.1. Quantification Frequency and Levels of Pesticides in Dust

Out of the 30 pesticides targeted, 10 were found in at least one dust sample (Table 1 and Appendix A, Figure 2). Chlorpyrifos and terbuthylazine were the most frequently quantified pesticides (96% and 91%, respectively), followed by carbaryl, diazinon, carbendazim, and tebuconazole (59–74%). The remaining four pesticides were rarely found (<10%) either only at Grabouw (i.e., diuron, malathion, and S-metolachlor) or at both sites (i.e., propiconazole) and are not further discussed. All homes had at least three pesticides in dust, and a maximum of eight pesticides was found for one household and one school, both located in Grabouw. The median dust concentrations measured in households of individual pesticides spanned several orders of magnitude and ranged from 4.38 ng g^−1^ (tebuconazole) to 365 ng g^−1^ (chlorpyrifos) (Table 1). Besides chlorpyrifos, only carbaryl and diazinon had dust concentrations higher than 1 µg g^−1^.

### 3.2. Temporal and Spatial Variations in Pesticide Levels in Dust

For those twelve households and one school for which dust samples were analyzed, on both day 1 and day 7, no significant temporal variations (*p* > 0.05, Mann–Whitney test) were observed for none of the pesticides investigated (Figure 3 and Appendix A). Indeed, for about two-third of all the pairs (*n* = 78) investigated, the dust concentrations of the individual pesticides measured on day 1 and day 7 were within 25% of each other, while large differences (up to a factor of 100) were found in the remaining cases (Figure 3 and Appendix A).

Significant differences in the levels of pesticides in dust were found between the areas (i.e., Hex River Valley and Grabouw) and the locations (village, farm, school). Indeed, when considering all samples (i.e., households, school, day 1, and day 7), the dust levels of carbaryl and carbendazim were significantly higher at Grabouw compared to Hex River Valley, while the opposite was found for tebuconazole (Appendix A). In addition, when focusing only on those samples collected on day 7, differences were observed between the different locations (farm, village, school) for some pesticides. Indeed, the concentrations of chlorpyrifos at Grabouw and tebuconazole at Hex River Valley on the farm were on average 6.16 and 2.22 times higher, respectively, than those in the village (Figure 4). On the other hand, at Grabouw, diazinon and tebuconazole had, respectively, 35.8 and 11.4 times higher levels in dust collected in village than those from the farm (Figure 4). Finally, at Grabouw, the two samples collected at the school had on average 12.2 times higher levels of carbendazim than those collected in households (farm or village).

### 3.3. Daily Intakes of Pesticides via Dust Ingestion

The daily intakes of individual pesticides via dust ingestion for children (estimated using the median concentrations measured in dust and the median ingestion rate) ranged from 0.16 (tebuconazole) to 100 (chlorpyrifos) pg kg^−1^ day^−1^ (Appendix A). The other cases considered (i.e., adults, maximum concentrations, and high ingestion rate) are presented in Appendix A and will be discussed only when the findings differ. Chlorpyrifos and carbaryl dominated the pesticide exposure via dust ingestion, as they contributed respectively for 64–76% and 11–22% at Hex River Valley and 19–50% and 43–68% at Grabouw, of the daily intakes (Figure 5). The total daily pesticides intakes of children were about three times higher in Grabouw compared to Hex River Valley when the median or maximum concentrations were used (Appendix A). Except for carbendazim at Grabouw, the children had daily intakes on average 4.4 times higher than adults. For carbendazim at Grabouw, the children-to-adult ratios of daily intake via dust ingestion were 22 and 38 for village and farm, respectively. When using the higher ingestion rate, the daily intakes of all individual pesticides were three times higher (Appendix A).

### 3.4. Comparison of Daily Intakes from Dust Ingestion with Inhalation and Soil Ingestion

In addition to dust, the same pesticides were also quantified in air and soil samples collected from this field campaign [68]. In total, 19 individual pesticides were found in at least one of these three environmental media investigated. Among these, six (i.e., acetochlor, alachlor, azinphos methyl, dimethachlor, malathion, and metazachlor) were found only in air, two (i.e., isoproturon and pirimicarb) only in soil, and one (i.e., diuron) only in dust. Therefore, their dominant routes of exposure were, respectively, inhalation, soil ingestion, and dust ingestion. For the remaining pesticides, the results discussed here are for the children using the median concentrations and ingestion rate (Figure 6), while those for adults, maximum concentrations and high ingestion rate are shown in Appendix A. For these ten pesticides that were found in at least two environmental matrices, six (i.e., atrazine, carbaryl, simazine, S-metolachlor, tebuconazole, and terbuthylazine) had inhalation as the major route (>90%) of exposure via the three studied pathways (i.e., inhalation, soil ingestion, and dust ingestion) at both sites. This was observed with all possible scenarios (Appendix A), except for carbaryl. On the other hand, for diazinon, it was dust ingestion (contributing for 62–94% depending on the site considered). For carbendazim, inhalation was the major route at Hex River Valley (>90%), but at Grabouw, it was dust ingestion (>97%). For propiconazole, inhalation was the major route at both sites, with dust ingestion being significant (37%) at Grabouw. For chlorpyrifos, inhalation dominated exposure at Hex River Valley (81% and 73% for the village and farm, respectively), while at Grabouw, there were pronounced differences between the two types of locations, with a contribution of inhalation, soil ingestion, and dust ingestion of 52%, 30%, and 18% at the village and 35%, 20%, and 45% at the farm. Using the high ingestion rate, the maximum concentrations or the input parameters for adults do not substantially modify the contribution of each exposure pathway for most of the pesticides investigated (Figure 6 and Appendix A). However, for carbaryl at Hex River Valley and tebuconazole at Grabouw, using the high dust ingestion rate or the maximum concentrations led to a significant increase of the contribution of dust ingestion, reaching about 20–30% of the overall daily intake.

In this study, all hazard quotients estimated using the daily intakes from the three exposure pathways were three to twelve orders of magnitude lower than one (Appendix A), suggesting minor risks. Carbaryl, chlorpyrifos, and tebuconazole were the compounds having the highest hazard quotients, up to 1.33 × 10^−3^ (Appendix A). Similarly, the cumulative exposures (data not shown) were at least three orders of magnitude lower than one, suggesting negligible risks.

## 4. Discussion

### 4.1. Quantification Frequency and Levels of Pesticides Found

In this work, ten pesticides were quantified in dust samples collected from thirty-eight households and two schools located in two agricultural areas in South Africa. The ratio of quantified-to-targeted pesticides in this study (i.e., 0.33) was smaller than in others done in the Northern Hemisphere (i.e., 0.50–1.00) [53,60,63,79]. The presence of pesticides in indoor dust is mainly affected by (i) the amount of pesticides applied in the vicinity (outdoors and indoors), (ii) the application technique used, (iii) the physico-chemical properties of individual pesticides and their degradation half-lives in air, dust and soil, and (iv) the meteorological conditions [42,80]. The low amount of pesticides quantified in this study could be related to the fact that pesticides are usually sprayed manually for these two crops (i.e., pome fruits and table grapes), which could limit the distribution of pesticides beyond the cropland in comparison to mechanical pesticide applications [81].

More specifically, chlorpyrifos, terbuthylazine, carbaryl, diazinon, carbendazim, and tebuconazole were frequently quantified (in 59–96% of the samples). In particular, chlorpyrifos, which is associated with neurotoxic and developmental effects [4,82], was the most frequently quantified pesticide in dust, similarly to what has been found in the USA [17,19,38,83], Pakistan [84], or Taiwan [81]. Its widespread occurrence in the two studied areas in environmental media (dust, air, soil, and water [64,68,69,70]) and human samples [64,82] is related to its common agricultural use, which was previously reported [70]. However, this pesticide, which is a candidate for the Stockholm Convention on Persistent Organic Pollutants [85], is prone to long-range atmospheric transport [86], and some fraction could have also been transported from other agricultural areas. Diazinon and carbaryl have also frequently been reported in other studies [17,38,79,87,88]. The presence of carbendazim and tebuconazole in dust samples has been only studied once from an agricultural region in the Netherlands, where they were also frequently found (>50%) [53].

The analysis of the presence of these pesticides in dust in comparison to air and soil that was previously reported [68] can provide valuable information. Among the 30 targeted pesticides, more were found in air (*n* = 16) [68] than in dust (*n* = 10) or in soil (*n* = 9) at these two sampling sites (Figure 2). In particular, carbaryl, chlorpyrifos, tebuconazole, and terbuthylazine were the pesticides found the most frequently in these three environmental matrices (Figure 2), which highlights their widespread occurrence in these two agricultural areas. Besides diuron, every pesticide found in dust was also present in air, and except diazinon (at both sites) and carbendazim (at Grabouw), their quantification frequencies in air were always higher than in dust. In addition, many pesticides never found in soils were often quantified (>20%) in air samples but never or rarely (<10%) in dust samples (e.g., malathion, propiconazole, S-metolachlor, acetochlor, alachlor, azinphos methyl, and dimethachlor). This could suggest some influence from medium- to long-range atmospheric transport and indicates that they do not penetrate the indoor environment or that the concentrations in dust were too low to be quantified. Atrazine and simazine, two relatively persistent triazines [78], were quantified only in soil and air but not in dust, which likely reflects their past agricultural use and volatilization from soils enhanced by higher temperatures [50]. Diuron was only quantified in dust in one household, which likely reflects its use at the domestic level. Overall, these results highlight the importance of air in the transport of pesticides from the outdoor environment to the indoors. Due to the significant correlations observed between the levels in dust and air of many organic compounds [79,89], several researchers have concluded that one matrix could be used as a surrogate for the other one using partitioning models [33,90]. This approach has been validated for many legacy pollutants [33,91,92] but showed rather poor performance for predicting the levels of chlorpyrifos in dust [79]. For pesticides, the lack of accurate data on their physico-chemical properties and the possible lack of equilibrium between dust and air for compounds that have a high octanol–air partitioning coefficient or that are currently used [33,79,93] limits our capacity to use these models [92]. However, as more pesticides were quantified in air than in dust, the validity of these partitioning models that were developed for more persistent substances is questionable.

The levels of individual pesticides reported here varied over several orders of magnitude (Table 1). For chlorpyrifos, for which the most data are available in the literature, the dust concentrations observed in this study were similar to those found in Taiwan [94], North Carolina [38], and Australia [89] but lower than in Pakistan [84] and higher than in Spain [95]. To the best of our knowledge, only one study done in the Netherlands investigated the levels of terbuthylazine, carbendazim, and tebuconazole in dust [53], in which terbuthylazine was rarely found, while the levels of carbendazim were higher, and those of tebuconazole were similar to the present study. Similar dust levels of diazinon (e.g., 10–200 ng g^−1^) were reported by previous studies [38,63,88].

### 4.2. Temporal and Spatial Differences in Pesticide Levels

For all pesticides investigated, no significant temporal differences in the dust levels measured at day 1 and day 7 were found for the 13 pairs investigated. This suggests that within short time scales (i.e., one week), one dust measurement provides a reliable estimate of human exposure. This finding is consistent with the findings from various field and laboratory studies. Indeed, the only two field studies existing on short temporal variations of pesticide levels in dust reported that within 5–8 days, measurements were relatively stable indicators of potential indoor exposure to pesticides [17,63]. Additionally, under laboratory conditions, the levels of cypermethrin and beta-cyfluthrin in dust samples remained constant for 56 days, with a significant decrease only observed 112 days after application [96]. This persistence in indoor dust could be due to limited solar radiation, constant indoor temperatures, lower microbial population, and moisture [21,47,96].

In terms of spatial variations, significant differences in the dust levels of some pesticides were found between the areas (i.e., Hex River Valley and Grabouw) and the locations (village, farm, school), which can help to identify the sources of these pesticides. Regarding the areas, three pesticides (i.e., carbaryl, carbendazim, and tebuconazole) showed significant differences in their dust levels, with carbaryl and carbendazim being the highest at Grabouw and tebuconazole at Hex River Valley. Interestingly, similar spatial variations were also found with soil and air samples for carbaryl and tebuconazole but not for carbendazim [68]. These consistent spatial variations observed in the three studied environmental matrices for carbaryl and tebuconazole suggests that agricultural activities control their environmental levels. On the other hand, significant differences between the two areas in the levels of terbuthylazine were found in air but not in soil or dust. This suggests that these three environmental matrices do not necessarily react in the same manner to point sources, further supported by the large spatial heterogeneity in levels of organic chemicals observed in soil [50] or dust [47].

Significant differences in pesticide dust levels were also found between the two locations (farm, village) studied. Indeed, the highest levels found in samples collected on farms for chlorpyrifos at Grabouw and for tebuconazole at Hex River Valley confirm that agricultural activities were the main source of pollution and that distance to agricultural fields is an important factor determining the levels of these pesticides in indoor dust. However, it is unclear whether these differences are due to spray drift and subsequent partitioning to dust, to take home exposures, or brought in via shoes, as it is known that indoor dust is composed of about 35% outdoor soil [97]. Negative associations between the levels of pesticides in house dust and the distance from the farms have also previously been reported in some studies [42,43,53,54,59,98] as well as positive ones between the levels of pesticides in house dust and agricultural acreages around the house [99], but this trend was not found in other studies [55,80,81]. In the present work, this was observed only for two pesticides out of the ten found. Therefore, further epidemiological studies should be cautious when using only geographic information systems (GIS) based on the distance to agricultural lands to predict human exposure to a large number of pesticides [43]. On the other hand, for diazinon and tebuconazole at Grabouw, the significantly higher levels found in dust collected from village compared to farm suggest these pesticides were used at the household level. Unfortunately, although questionnaires were deployed in this study [20], they failed to identify the specific active ingredients used at the domestic household level. Several studies have also shown the importance of residential use of pesticides on their dust levels for imidacloprid, malathion, chlorpyrifos, diazinon, carbaryl, or prallethrin [53,54,57,59,63,81,87]. It is interesting to note that tebuconazole showed a distinct behavior between the two sites, which shows that generalization about the influence of proximity to agricultural fields on the levels of pesticides in dust is not accurate, and site-specific differences should be considered in further exposure assessment to pesticides. Finally, at Grabouw, the significantly higher levels of carbendazim found at the school, for which existing data are scarce [44,58], compared to the households suggests its use in the close vicinity or within the school during the sampling campaign.

### 4.3. Daily Uptakes of Pesticides via Dust Ingestion

Chlorpyrifos and carbaryl, which are known for their toxic effects on humans [4,100], had the highest daily intakes at both sites showing the importance of these two pesticides in terms of human exposure for the agricultural residents of these two sites. Except for carbendazim, children had daily intakes of pesticides about four times higher than adults. This is similar to previous studies [89,95] and is due to their higher ingestion rate and lower weight (Appendix A). Considering that young children are a particularly vulnerable group due to the well-documented detrimental effects of pesticide exposure on child neurodevelopment [101,102], the higher exposure levels reported in this and other studies require further attention. For carbendazim, the children-to-adult ratio of daily intake via dust ingestion was much higher (i.e., 22–38 at Grabouw) due to the significantly higher levels of carbendazim measured in the two dust samples collected at schools where adults are not exposed. This highlights the importance of having several micro-environments investigated when assessing human exposure to organic chemicals via dust ingestion, particularly for children. In addition to the micro-environment frequented, the dust ingestion rate is an important factor that can lead to uncertainties within a factor of three and should therefore be better characterized.

### 4.4. Comparison of Daily Intakes from Dust Ingestion with Inhalation and Soil Ingestion

Daily uptakes of pesticides via dust ingestion were compared with those from inhalation and soil ingestion previously reported [68]. Inhalation was generally the major route of exposure (>90%) for 13 out of the 19 pesticides quantified in at least one of the three environmental matrices investigated. For the remaining six pesticides, the major pathway was soil ingestion for isoproturon and pirimicarb and dust ingestion for diuron and diazinon, while chlorpyrifos and carbendazim differed depending on the site considered (Figure 6). Considering that the air concentrations were obtained from the outdoor environment where concentrations of organic compounds such as pesticides are several orders of magnitude lower than indoors [92], this highlights the importance of inhalation in non-dietary exposure to pesticides. This is similar to what has been found for North Carolina children, who were about 10 times more exposed to chlorpyrifos via inhalation than via both soil and dust ingestion [38]. Similarly, it was estimated that dust was contributing less than 15% of the levels of dialkyphosphates metabolites measured in urine [55]. One can notice the important role of input parameters such as the ingestion rate or the concentrations used (median or maximum) in the contribution of dust ingestion for carbaryl and tebuconazole, which increased by a factor of three. For other semi-volatile organic compounds, several studies have found that uptake via inhalation was higher than via dust ingestion and dermal contact with dust for compounds that are volatile, while the opposite was found for the non-volatile compounds [36,79]. However, this influence of physico-chemical properties on the contribution of different exposure pathways was not observed in our study.

The health risks estimated in this study were negligible, both for individual pesticides or cumulative exposure. This was also found by several other studies (focusing only on dust ingestion) [63,89,95]. However, we should keep in mind that the risks estimated in this study from several environmental matrices do not take into account dietary ingestion, which can dominate human exposure [38,103,104,105]. Carbaryl, chlorpyrifos, and terbuthylazine, which were the pesticides the most frequently found in all the three matrices investigated, had the highest hazard quotients. Therefore, further studies could investigate the health risks due to their transformation products, as many of them could have significant health effects [106].

### 4.5. Limitations and Strengths

The main strengths of this study are: (i) the characterization of the levels of several herbicides and fungicides that have been poorly characterized worldwide in dust, (ii) the first characterization of daily intakes via dust ingestion of multiple pesticides in Africa, and (iii) the improvement of our understanding on the contribution of several non-dietary exposure pathways. This study has three main limitations. The first concerns the characteristics of the sampling campaign, which was short (seven days at each site) and involved a limited amount of samples (*n* = 54). In addition, dust is associated with large spatial heterogeneity of the levels of contaminants [92,107], particularly before and after pesticide use [39]. Therefore, the spot samples collected might not be representative of the entire room. Secondly, there are high uncertainties associated with the input parameters of the model used to characterize human exposure via the three pathways. Indeed, the data on the bioaccessibility of pesticides or dust ingestion rates are highly uncertain [47], which could contribute to a 20-fold variability in the daily doses estimates [108]. In addition, outdoor and not indoor air levels of pesticides were used. However, we expect this to have a minor effect, as several studies have found a significant correlation between these [19,98], and levels indoors are usually higher than outdoors [92,109,110]. Finally, the outcome of the health impact assessment is limited by the fact that it considers only a limited amount of active ingredients and does not take into account synergistic effects [111]. Additionally, some mechanisms of toxicity (e.g., suppressed expression of serotonin transporter genes) are not considered in the definition of the reference dose [63].

## 5. Conclusions

In this study, the spatial and temporal variations of 30 pesticides in dust and the human exposure via dust ingestion in comparison to inhalation and soil ingestion were investigated at two agricultural sites in South Africa. Within seven days, no significant temporal variations in the dust levels of individual pesticides were found. On the other hand, significant spatial variations were observed for some pesticides, highlighting either the importance of proximity to agricultural fields (chlorpyrifos at Grabouw and tebuconazole at Hex River Valley) or use at the domestic level (diazinon at Grabouw and tebuconazole at Grabouw) or applied at or in the vicinity of the school (carbendazim at Grabouw). The agricultural residents of the two sites investigated were exposed to 10 pesticides via dust ingestion. However, this exposure pathway was found negligible (<10%) compared to inhalation or soil ingestion for 14 out of the 19 pesticides found in dust, air, or soil. Further studies should confirm this finding by characterizing the levels of pesticides in several environmental matrices and the contribution of several non-dietary exposure pathways in both spraying and non-spraying seasons.

## Figures and Tables

**Figure 1 toxics-10-00629-f001:**
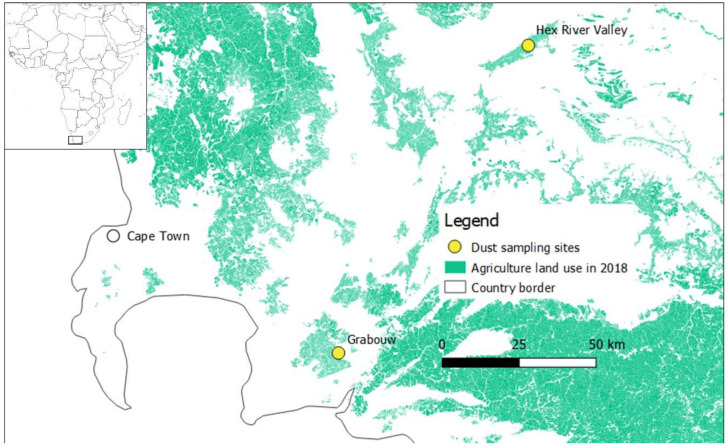
Map of the sampling sites.

**Figure 2 toxics-10-00629-f002:**
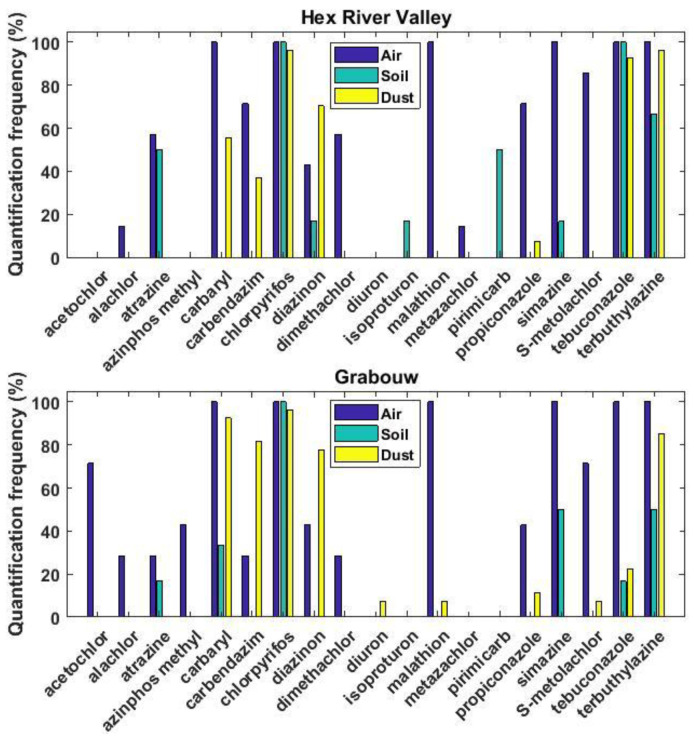
Quantification frequencies of the individual pesticides in air, soil, and dust. Data on pesticides in air and soil were obtained from [68].

**Figure 3 toxics-10-00629-f003:**
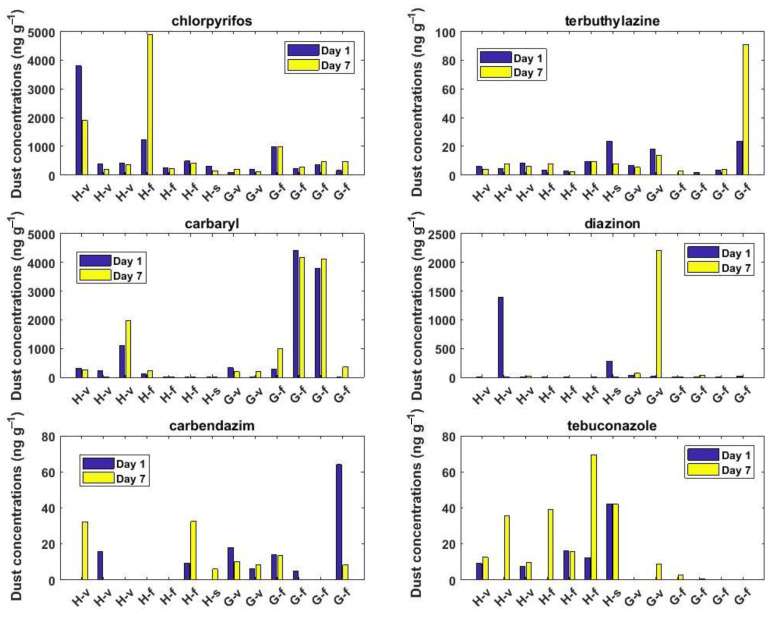
Temporal variations of pesticide levels in dust samples (in ng g^−1^) collected at Hex River Valley (H) or Grabouw (G) at households living in farms (f), village (v), or at the school (s).

**Figure 4 toxics-10-00629-f004:**
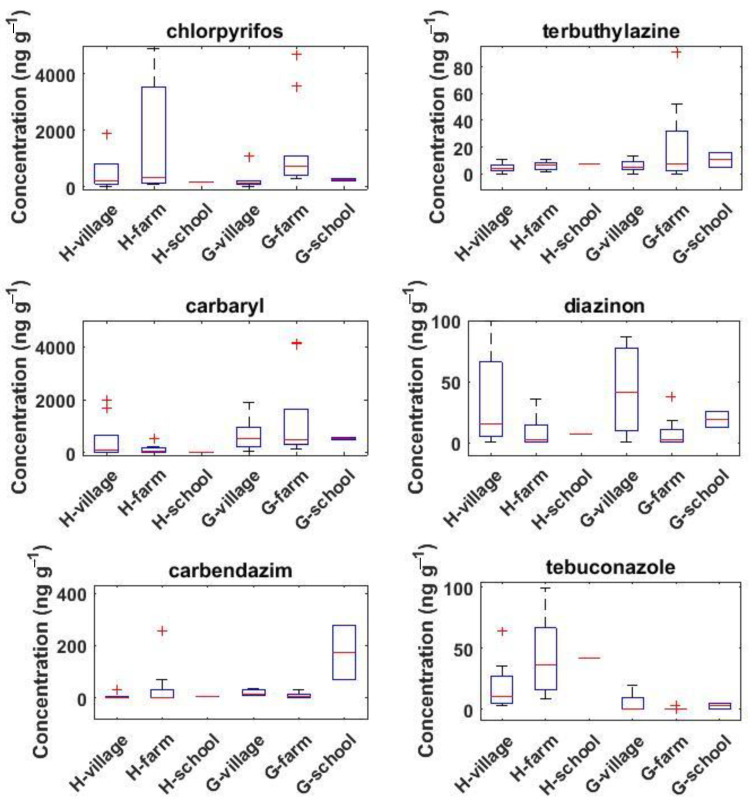
Spatial variations in dust levels of pesticides (in ng g^−1^) among the different sites and areas from the samples collected on day 7 only (*n* = 41). H and G denote Hex River Valley and Grabouw, respectively. Some outliers are not shown for better visibility. Boxplots represent the 25–75th percentile, whiskers represent the minimum and maximum values (excluding outliers which are shown as the red crosses) and the line within the box represents the median value.

**Figure 5 toxics-10-00629-f005:**
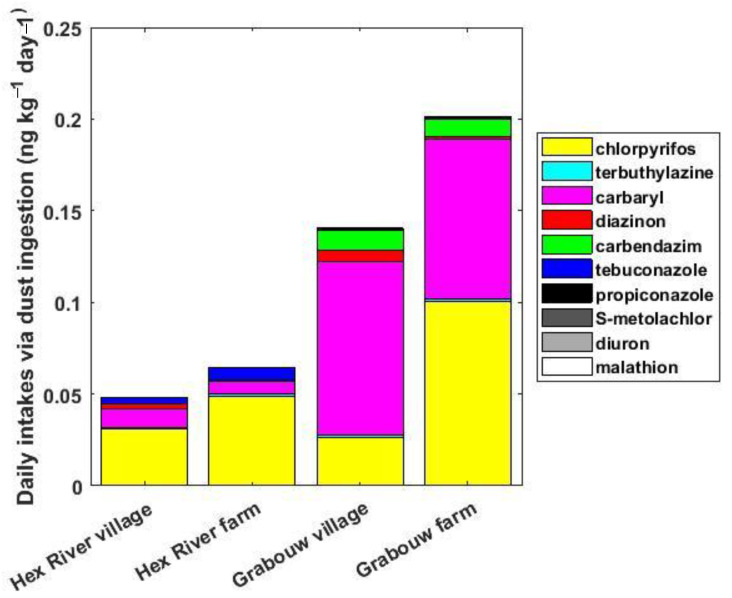
Daily intake of pesticides (in pg kg^−1^ day^−1^) via dust ingestion for children using the median concentrations measured and the median ingestion rate.

**Figure 6 toxics-10-00629-f006:**
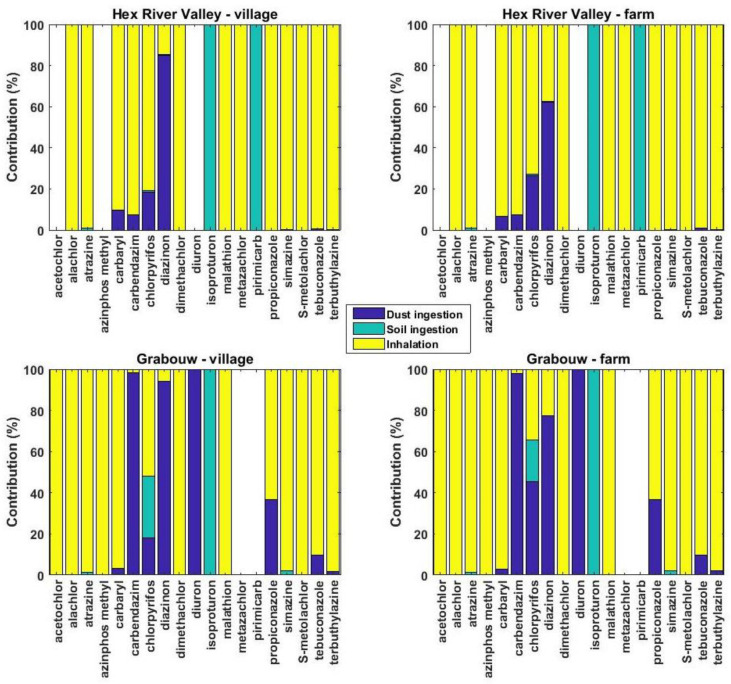
Contribution of three exposure pathways (dust ingestion, soil ingestion, and inhalation) on the daily uptake of pesticides of children living at farm and village locations at Hex River Valley and Grabouw using the median concentrations. Blank columns corresponds to the cases when a pesticide was not quantified in air, soil, and dust.

**Table 1 toxics-10-00629-t001:** Summary of the concentrations (in ng g^−1^) of individual pesticides found in the dust samples collected in households. QF indicates quantification frequency in percentage, IQR indicates interquartile range. For propiconazole, S-metolachlor, diuron, and malathion, only the concentrations above the quantification limits were considered, while for the remaining pesticides, imputed data were considered for statistics.

		Chlorpyrifos	Terbuthylazine	Carbaryl	Diazinon	Carbendazim	Tebuconazole	Propiconazole	S-Metolachlor	Diuron	Malathion
All *n* = 50	QF	96	90	76	72	60	58	8	4	2	4
	Mean	1250	9.30	1020	122	16.3	13.6				
	Median	365	4.54	247	9.29	7.02	4.38				
	Min	0.19	0.05	5	0.29	0.14	0.19	3.63	10.3	26.8	43.9
	Max	19,500	90.8	17,200	2210	257	99.0	12.5	46.6	26.8	150
	IQR25	135	2.76	64	0.30	0.15	0.20				
	IQR75	986	9.25	544	31.2	14.2	15.8				
Hex River Valley *n* = 25	QF	96	96	60	68	36	92	8	0	0	0
	Mean	1810	5.38	292	136	17.5	25.1				
	Median	398	4.47	102	6.41	0.15	15.7				
	Min	0.19	0.05	5	0.29	0.14	0.20	3.63			
	Max	19,500	11.0	1980	1680	257	99.0	9.60			
	IQR25	142	3.04	5	0.29	0.15	7.96				
	IQR75	1850	7.70	266	15.5	7.45	35.4				
Grabouw *n* = 25	QF	96	84	92	76	84	24	8	8	4	8
	Mean	690	13.2	1740	109	15.1	2.04				
	Median	268	4.62	525	11.2	11.0	0.20				
	Min	0.20	0.05	5	0.29	0.14	0.19	3.63	10.3	26.8	43.9
	Max	4700	90.8	17,200	2210	64.1	19.6	12.5	46.6	26.8	150
	IQR25	133	2.70	207	4.37	5.94	0.19				
	IQR75	948	13.6	1660	37.4	17.8	0.20				

## Data Availability

Data available on request.

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
