# Peer review of "Human Exposure to Pesticides in Dust from Two Agricultural Sites in South Africa"

_toxics, 2022, doi:10.3390/toxics10100629_

Round 1

Reviewer 1 Report

General comment:

The manuscript brings useful information about the spatial and temporal variations of pesticides in dust at two agricultural sites in South Africa. It also assessed human exposure to pesticides via dust ingestion in comparison to inhalation and soil ingestion. The results are interesting, the authors found the most common pesticides are chlorpyrifos, terbuthylazine, carbaryl, diazinon, carbendazim and tebuconazole. They did not find any significant temporal variations, but they find significant spatial variations for some pesticides. The reviewer believed that the present study is interesting and potentially could contribute to the research field, however, there are some concerns and questions that require be addressed to clarity and improve the present version.

Specific comments:

1.     Title- the title is informative and represents the major findings. However, authors may reconsider using the words/phrase “current use” as some of the pesticides are persistent in the environment.

2.     Abstract- in the abstract, the aim of the study is clearly mentioned, and major results are also properly presented. Please mention the names of four pesticides that were quantified in dust and human intake via dust ingestion was more than 10% compared to inhalation or soil ingestion. It would be better if the authors could add their recommendations/suggestion at the end of the abstract.

3.     Introduction-The introduction is well-written and well-discussed. However, the authors could discuss more the use of pesticides, the most commonly used pesticides, amounts of pesticides used in South Africa yearly and so on.

4.     Materials and Methods- Authors should add a map/figure of sampling sites in the materials and methods sections.

5.     Results- the quality of the figures is not satisfactory. More high-resolution figures should be added. Please compare your results with other published reports.

6.     Discussion- the discussion section could be more focused and specific. Authors can give more emphasis to discussing their findings from multiple angles.

7.     Conclusion- the conclusion does not properly answer the aims of the study (i.e., human exposure). Opportunities to inform future research are not addressed properly.  

Reviewer 2 Report

In a study performed in South Africa, Degrendele et al. investigated the spatial and temporal variations of 30 pesticides in dust and estimated human exposure via dust ingestion. This is a neglected and interesting research topic. This study was well performed and has good quality. My comments and some suggestions are detailed below:

- Title: I suggest simplifying it to “Human exposure to pesticides in dust from two 2 agricultural sites in South Africa”.

- Abstract: It is appropriated. No change is needed.

- Introduction, line 39 e 40: The authors state “and have therefore been classified as emerging contaminants with the potential to adversely affect human health”. I suggest removing this part from the paragraph since pesticides are fully recognized contaminants and solid (not potential) evidence supports the detrimental impact of pesticides on human health.

- Introduction, first paragraph: Please add some information concerning the impact of pesticides on DNA damage in human populations.

- Introduction, line 75: Change to “against insects (e.g. mosquitoes, fleas, ticks)”.

- Materials and Methods: Please, put Figure S1 as a conventional figure in the main article body. It will help to give (geographical) context to the study.

- Materials and Methods, line 154: Please, close the parenthesis.

- Materials and Methods, Data analysis: Describe in detail the software (or packages) used to perform the analyses and plot the graphs (same for Figure S1).

- Results: No change is needed.

- Discussion, subsection 4.3 Daily uptakes of pesticides via dust ingestion: I suggest including more information concerning the detrimental effects of pesticides on child development, evidencing with references why exposure in childhood is more problematic than pesticide exposure in adults.

- Conclusion: No change is needed.

- Language: English is fine.

Reviewer 3 Report

Dear Authors,

Congratulations for your work. The research described on the manuscript is extensive. Therefore, the manuscript is long, detailed and not easy to be fully understood. 

However, I find that some figures that are included as supplementary material should be included inside the manuscript main body, making it even longer. 

Please address the following comments and suggestions to improve your manuscript before final publication:

  • The location map should not be included on the supplementary information set but inside the manuscript. 
  • Add on Section 2.1 a detailed map with the location of all the sampling sites and relevant information to understand the collection process of dust samples. 
  • Section 2.2. Why was this methodology used? Were any laboratory standards being used? Please explain. 
  • Figures S2 to S5 should be included inside the manuscript when they are cited, and not as supplementary material. Figure S6 to S10 shall remain as supplementary material. 
  • Please add the following three references on Lines 337 and 343, to refer to previous studies that identified terbuthylazine and chlorpyrifos in soil and water coming from common agricultural uses on other locations: 

Rodrigo-Ilarri, J.; Rodrigo-Clavero, M.-E.; Cassiraga, E.; Ballesteros-Almonacid, L. Assessment of Groundwater Contamination by Terbuthylazine Using Vadose Zone Numerical Models. Case Study of Valencia Province (Spain). Int. J. Environ. Res. Public Health 2020, 17, 3280. https://doi.org/10.3390/ijerph17093280

Pérez-Indoval, R.; Rodrigo-Ilarri, J.; Cassiraga, E.; Rodrigo-Clavero, M.-E. Numerical Modeling of Groundwater Pollution by Chlorpyrifos, Bromacil and Terbuthylazine. Application to the Buñol-Cheste Aquifer (Spain). Int. J. Environ. Res. Public Health 2021, 18, 3511. https://doi.org/10.3390/ijerph18073511

Ricardo Pérez-Indoval, Javier Rodrigo-Ilarri, Eduardo Cassiraga, María-Elena Rodrigo-Clavero, PWC-based evaluation of groundwater pesticide pollution in the Júcar River Basin, Science of The Total Environment,

Volume 847, 2022, 157386, ISSN 0048-9697, https://doi.org/10.1016/j.scitotenv.2022.157386

Round 2

Reviewer 1 Report

No more comments. However, it would be better if the authors carefully read the revised manuscript several times during proofreading.